# Development and evaluation of the psychometric properties of a brief parenting scale (PS-7) for the parents of adolescents

**Sai-fu Fung**[ORCID]*, **Annis Lai Chu Fung**

Department of Social and Behavioural Sciences, City University of Hong Kong, Kowloon, Hong Kong

* sffung@cityu.edu.hk

## Abstract

This study aimed to develop a seven-item brief parenting scale (PS-7) based on the original parenting scale (PS) and various other shortened versions and with a better factor structure for the parents of adolescents. The scale was tested with a sample of 3,777 parents (2,205 mothers and 1,572 fathers). Confirmatory factor analysis was used to evaluate the dimensionality of the different versions of the PS. Only the PS-7 with a two-factor structure, i.e., laxness (three items) and overreactivity (four items), showed a good model fit based on a representative sample of parents of junior secondary school students. Overall, the results suggest that PS-7 is comparable to the original PS and possesses good psychometric properties in terms of internal consistency, factorial validity, construct validity, criterion validity and discriminant validity. The abbreviated parenting scale also provides a reliable and cost-effective method for assessing parental practices for treatment and assessing treatment outcomes.

**Data Availability Statement:** All relevant data that will enable future readers to replicate our findings are uploaded as a Supporting Information file.

**Funding:** This work was supported by the Research Grant Council, Hong Kong under General

## Background

Parenting has been shown to have an important influence on the mental well-being of children and adolescents and the prevalence of behavioural problems [1–4]. The literature has also suggested that parenting influences children's school performance and has linked parenting styles to controversies arising over cultural differences. Asian, and especially Chinese, parenting styles have been categorised as controlling and authoritarian, and are popularly known as 'tiger' parenting [5]. However, a longitudinal study of 444 Chinese American families that examined the effects of parenting styles on adolescent adjustment suggested that tiger parenting was not the most typical profile [6]. Going beyond the common perception of Asian parenting as controlling and authoritarian [7–10], recent studies have suggested that close parental control and an authoritative parenting style are fused with ideas of training and presence that help to explain school achievement [11, 12]. Parenting style has now been accepted as a cross-cultural concept that enhances understanding of child behaviour across ages and ethnicities [7, 13, 14]. Hence, developing a convenient cross-cultural measure of parenting styles is highly warranted for epistemological research.

Research Fund [number 11611517]. The funder had no role in study design, data collection and analysis, decision to publish, or preparation of the manuscript.

To operationalise parenting style, Arnold, O'Leary [15] developed a parenting scale (PS) based on a sample of 168 mothers of children ranging from 18 to 48 months old (98 boys and 70 girls). Subsequent studies suggested that the scale was applicable not only to mothers of toddlers, but also to parents of both genders with children and young adolescents attending primary and secondary schools [16–21]. The PS has since been widely accepted internationally as a measure of parenting behaviour [16]. The original and adapted versions of the scale have been translated into numerous languages, including Chinese [17], Dutch [18], French [22], German [16, 19], Japanese [23, 24], Persian [19], Spanish [25], Swedish [20] and Vietnamese [21]. The scale has also been used to examine the behaviour of parents in different contexts, such as community-based paediatric practices for routine care in America [26], Australian mothers with preschool-aged children [27], parents of school-aged children with ADHD [28] and clinical populations [29].

Nevertheless, several factors may limit the application of the full version of the PS. The scale originally comprised 30 items with a three-factor structure, comprising laxness (11 items), overreactivity (10 items) and verbosity (7 items; with two multi-factor items, 7 and 9). The scale developers reported that four items (1, 5, 13 and 27) with low factor loading values (below 0.35) were categorised as not loading on a specific factor and were excluded from the scale. Hence, the 26 item PS with a three-factor structure is commonly used. The original scale was derived based on exploratory factor analysis (EFA), and ambiguous results were obtained for the dimensionality and number of items per factor. In particular, verbosity was found to have a complicated factor structure and coefficients with a questionable alpha value of 0.63 [15]. Moreover, the numerous studies conducted during the early development and application of the scale mainly focused on relatively small samples of mothers with infants and English-speaking populations [18, 30].

To address these issues, many early studies attempted to provide a shortened version of the PS [31]. However, these studies used limited validation tools to evaluate the latent structure of the scale, such as EFA to uncover the underlying structure or confirmatory factor analysis (CFA) to verify the factor structure [20]. The following five brief versions are the most significant examples and have been widely used in the field. Salari, Terreros [20] proposed a 21-item scale (PS-21) in which all of the verbosity items were removed and the original two sub-scales, laxness (11 items) and overreactivity (10 items), were included after evaluating the psychometric properties of the scale. However, the CFA failed to fulfil the goodness-of-fit indices, i.e., chi-square divided by less than or equal to three degrees of freedom or a comparative fit index (CFI) higher than 0.950 [32–34]. One of the original PS scale developers, Susan O'Leary, and her colleague proposed a 13-item shortened version of the scale (PS-13) with a three-factor structure comprising laxness (five items), overreactivity (five items) and hostility (three items) [30]. However, their newly proposed factor, hostility, had a problematic Cronbach's alpha value of 0.52. Irvine, Biglan [35] developed a version of the PS for adolescents (PS-12), based on a sample of 298 parents (94.5% mothers) of school students who identified as being at risk for problem behaviour. The 12 items were derived from the original PS sub-scales for laxness (six items) and overreactivity (six items). Intriguingly, without the support of EFA or CFA, the authors further suggested adding an additional single monitoring item, i.e., item 13, which had been removed from the original PS scale due to low factor loading. Another shortened version was based on the findings of two studies on 187 and 216 American mothers, which suggested using a 10-item PS (PS-10) with a two-factor structure comprising laxness (5 items) and overreactivity (5 items) [31]. Nevertheless, because the studies focused solely on mothers, the results may have limited applicability to fathers. Finally, the latest attempt was the eight-item parenting scale short form (PS-8), which comprised two sub-scales, laxness (four items) and overreactivity (four items), derived from a sample of 539 German parents (312 mothers and

227 fathers) of children aged from 1 to 18 [16]. Although the results were convincing, further tests and evaluations are needed to assess its generalisability to other contexts. The items and factor structure of these PSs are summarised in the S1 Appendix.

This study has two main aims. First, to evaluate the factor structure of the full PS and variants of the shortened versions using CFA and a larger sample comprising the parents (both fathers and mothers) of adolescents. Second, to propose a seven-item brief parenting scale (PS-7) that has a better factor structure and better psychometric properties than the existing versions.

## Methods

This study was approved by the ethical committee of the City University of Hong Kong. Its procedure was in compliance with the Declaration of Helsinki guidelines. All of the participants gave informed consent prior to the study.

### Participants and measures

In January 2018, 4,007 respondents from 10 secondary schools located in different districts of Hong Kong were recruited to participate in this cross-sectional study. Respondents who were either the father or mother of an adolescent were included in the analysis (N = 3,777). The valid sample consisted of 2,205 mothers and 1,572 fathers (average age 44.83 years; SD = 6.95) of junior secondary school students (i.e., Forms 1 to 3) aged between 12 to 14 [17]. The demographic information of the participants is summarised in Table 1. The unique historical context of Hong Kong, with its mix of Eastern and Western cultures, provides an ideal research setting for investigating parenting styles because it may generate results that are relevant not only to Chinese society, but also to other Anglo-Saxon societies [36–38].

The full PS consists of 26 items with a three-factor structure comprising 11 items related to laxness (7, 8, 12, 15, 16, 19, 20, 21, 24, 26 and 30), 10 items for overreactivity (3, 6, 9, 10, 14, 17, 18, 22, 25 and 28) and 7 items for verbosity (2, 4, 7, 9, 11, 23 and 29). There are two multi-factor items: item 7, which is related to both laxness and verbosity, and item 9, which is associated with overreactivity and verbosity. The parents rated the items on a 7-point Likert scale to indicate their tendency to use specific strategies to discipline their children [15]. The scale items were translated into Chinese using the back-translation procedure by two bilingual translators who were familiar with both Chinese and English and were fully aware of the issues and techniques relating to cross-cultural research [39–41].

### Item selection process

The process is based on the criteria, the latest practice and recommendations used in the existing PS studies [16, 18] and other scale development and validation literature [42–49]. The selected items have gone through the following two-step procedure. Step one, selecting the items: i) using inductive approach to analyze the correlation matrix of all the items and keeping the items with 0.250 or above. We also cross-checking the Cronbach's alpha, if deleted and McDonald's omega values to ensure that the shortened version is above the acceptable range > 0.70; ii) using scree test in factor analysis to identify the factor structure with eigenvalues higher than 1.0 [50]. We also select the items with highest factor loadings, i.e. > 0.50 and avoid items involve correlating the error terms based on the modification indices. When selecting the items, we try to retain the sufficient items (at least three) in each factor to ensure that the validity standard of the shortened version is equivalent to the full version; iii) to verify the abbreviated version with the confirmatory factor analysis to ensure that the scale with good construct validity, i.e. fulfil all the stringent requirements for good model fit. Step two,

**Table 1. Participant demographic characteristics.**

| Variable | Respondents |
|---|---|
| Filler's age mean (SD) | 44.83 (6.95) |
| Partner's age mean (SD) | 45.35 (7.03) |
| Relationship with the target child n (%) | |
| Mother | 2,205 (55%) |
| Father | 1,572 (39.2%) |
| Others | 160 (3.9%) |
| Missing | 70 (1.8%) |
| Children school year | |
| Form 1 n (%) | 1,110 (27.7%) |
| Form 2 n (%) | 1,150 (38.7%) |
| Form 3 n (%) | 1,347 (33.6%) |
| Number of children (SD) | 2.26 (0.98) |
| Education level n (%) | |
| No formal education | 973 (24.3%) |
| Primary education | 1,520 (37.9%) |
| Secondary education | 1,096 (29.0%) |
| Diploma or college | 58 (2.0%) |
| Tertiary education | 116 (2.9%) |
| Missing | 155 (3.9%) |
| Martial status n (%) | |
| Single | 46 (1.1%) |
| Married | 3,295 (82.2%) |
| Divorce/separated | 288 (7.2%) |
| Cohabit | 106 (2.6%) |
| Widowed | 107 (2.7%) |
| Missing | 165 (4.1%) |

ensuring that the compatibility between the full scale and shortened version: iv) we adopted the following practice of Kliem, Lohmann [16], 'short form should also correlate strongly with the original PS on the total score level as well as on the subscale (overreactivity and laxness) level' (p. 34). As such, there should be significant strong positive correlation ($> 0.80$) between the full and short scales, including their sub-scales; and v) lastly, the abbreviated version should possessing good criterion validity as reported in the existing PS literature.

## Procedure

The sample (N = 3,777) was randomly stratified into three datasets (samples 1, 2 and 3). Each sub-sample consisted of 1,259 cases that reflected the original sex ratio of the participants, i.e., mothers 58.4% and fathers 41.6%, to avoid the problem of overfitting when using EFA and CFA to evaluate the factorial and construct validity of the scale [51, 52].

 Various psychometric testing tools and validated instruments were used to examine the newly proposed PS-7. EFA was used to evaluate the factorial validity and the principal axis method with oblique rotation was used to evaluate the factor structure of the scale [18, 34, 53]. In addition, the Kaiser-Meyer-Olkin (KMO) test and Bartlett's test of sphericity were used to evaluate the model sufficiency. The KMO estimates were over 0.70 and the Bartlett's test was significant ($p < 0.01$), thus indicating that the scale had a satisfactory factor structure [54]. According to Hair [34], an item with a factor loading over 0.50 is regarded as having practical

significance in studies with over 350 respondents. The internal consistency of the scale was assessed by Cronbach's alpha [55], McDonald's omega [56–58] and the corrected item-total correlation between the seven items [34, 59].

CFA was used to replicate and evaluate the construct validity of the scales [42, 60, 61]. Diagonally weighted least squares (DWLS) was used as the CFA estimator to examine the factor structure of the PS for two reasons. First, the literature suggests that the PS has high item-level skewness and kurtosis [30]. Second, because scales with latent constructs estimated by Likert scale items consist of ordinal data, DWLS is regarded as the least biased and most optimal fit [62–66]. The model fit and cut-off criteria were evaluated on the basis of the values suggested in the structural equation modelling (SEM) literature. Specifically, over 0.950 for both CFI and the Tucker-Lewis fit index (TLI), below 0.08 for the standardised root mean square residual (SRMR) and below 0.06 for the root mean square error of approximation (RMSEA) are considered to indicate a good fit [32, 34, 67, 68]. In addition, model acceptability was indicated by $\chi^2$ / df $\leq$ 3 due to the large sample size [33, 69].

The criterion validity was evaluated using other validation constructs and measurements reported in the literature on parenting. The PS has been reported to be significantly positively related to aggressive and delinquent behaviours [30, 31, 35], authoritative parenting [31], ADHD and cognitive and hyperactivity symptoms [30]. Hence, the following well-established scales were used to evaluate the criterion validity of PS-7. The reactive–proactive aggression questionnaire (RPQ) comprises 23 items to measure reactive (11 items) and proactive (12 items) forms of aggression on a Likert-type scale ranging from 0 = *never* to 2 = *usually* [70–72]. The child behaviour checklist (CBC) consists of 33 items identifying aggressive (20 items) and delinquent (13 items) behaviours on a 3-point Likert-type scale ranging from 0 = *unsuitable* to 2 = *very suitable* [73–76]. Conners' parent rating scale (CPRS) comprises 28 items with a 4-point Likert-type scale (0 = *never*; 4 = *a lot*) for parents to rate their child in four dimensions, namely ADHD, oppositional, cognitive problems and hyperactivity [77, 78]. The parenting styles and dimensions questionnaire (PSDQ) is evaluated on a 5-point Likert-type scale (1 = *never*; 5 = *always*), with a particular focus on the three dimensions of physical coercion (five items), punitive (three items) and verbal hostility (three items) [13, 79].

In addition, Reitman, Currier [31] found that the original PS was not correlated with the educational level of the parent, and this study attempted to replicate this finding to demonstrate the discriminant validity of PS-7 [80]. The above analyses were all implemented with IBM SPSS 25.0 and the lavaan package version 0.6–3 [81] in R computing environment 3.6.0.

## Results

### Development of the seven-item brief parenting scale using EFA

The seven-item parenting scale was inspired by PS-12 [35], PS-10 [31] and PS-8 [16]. The selection of items for the brief version adhered to the existing practices recommended in the scale development and validation literature, with a particular focus on the cultural context and the results of inter-item correlations, corrected item-total correlations, Cronbach's alpha, McDonald's omega and EFA [42, 51]. The detail item selection procedure and criteria have been stated in the methods section. According to the results, the newly proposed PS-7 has a two-factor structure comprising laxness (items 16, 20 and 30) and overreactivity (items 6, 10, 14 and 17) (see the S1 Appendix). The KMO test (0.823) and Bartlett's test of sphericity ($\chi^2$ = 2452.585, $p < .001$) factor analysis results from sample 1 (n = 1,259) indicate that PS-7 has an appropriate scale construction. The EFA results using the oblique rotation method (Table 2)

**Table 2. Factor loading results from exploratory factor analysis of PS-7.**

| Item | Laxness | Overreactivity |
|---|---|---|
| 16. When my child does something I don't like, I often let it go | **0.857** | 0.381 |
| 20. When I give a fair threat or warning, I often don't carry it out | **0.847** | 0.380 |
| 30. If my child gets upset, I back down and give in. | **0.733** | 0.366 |
| 6. When my child misbehaves, I usually get into a long argument with my child. | 0.333 | **0.775** |
| 10. When my child misbehaves, I raise my voice or yell. | 0.415 | **0.801** |
| 14. After there's been a problem with my child, I often hold a grudge. | 0.340 | **0.747** |
| 17. When there's a problem with my child, things build up and I do things I don't mean to do. | 0.341 | **0.747** |

suggest that the two factors extracted with eigenvalues greater than 1.0 (3.123 for items related to laxness and 1.172 for items related to overreactivity) from PS-7 account for 62.195% of the total variance. The items related to laxness explain 45.708% of the variance, with factor loadings ranging from 0.733 to 0.857. The overreactivity items, which have factor loadings ranging from 0.747 to 0.801, explain 16.487% of the variance. The EFA results replicate the latent structure of the two factors, namely laxness and overreactivity, as suggested in the PS literature [15, 18, 35].

## Internal consistency

Table 3 presents the descriptive statistics and item correlations for all seven items of PS-7 from sample 1. The corrected item-to-total correlations for PS-7 range from 0.470 to 0.599, which is similar to the range of 0.42 to 0.65 reported by Kliem, Lohmann [16]. Cronbach's alpha for the seven-item PS (0.799) is comparable to that reported by Kliem, Lohmann [16] (0.75) and to the values reported in other related studies. McDonald's omega (0.83) also suggests that PS-7 has good internal consistency.

**Table 3. Descriptive statistics and items correlations for the 7-item parenting scale items.**

| Item | 16 | 20 | 30 | 6 | 10 | 14 | 17 |
|---|---|---|---|---|---|---|---|
| 16 | 1.000 | 0.441*** | 0.632*** | 0.300*** | 0.301*** | 0.355*** | 0.363*** |
| 20 | 0.422*** | 1.000 | 0.469*** | 0.297*** | 0.284*** | 0.258*** | 0.331*** |
| 30 | 0.613*** | 0.441*** | 1.000 | 0.305*** | 0.300*** | 0.299*** | 0.355*** |
| 6 | 0.263*** | 0.278*** | 0.290*** | 1.000 | 0.445*** | 0.438*** | 0.549*** |
| 10 | 0.283*** | 0.273*** | 0.276*** | 0.423*** | 1.000 | 0.447*** | 0.487*** |
| 14 | 0.312*** | 0.238*** | 0.279*** | 0.421*** | 0.417*** | 1.000 | 0.490*** |
| 17 | 0.330*** | 0.312*** | 0.338*** | 0.518*** | 0.466*** | 0.471*** | 1.000 |
| Mean | 2.50 | 3.12 | 2.86 | 2.83 | 2.78 | 1.92 | 2.74 |
| SD | 1.256 | 1.539 | 1.397 | 1.402 | 1.498 | 1.217 | 1.405 |
| Skewness | 0.579 | 0.302 | 0.284 | 0.311 | 0.370 | 1.260 | 0.287 |
| Kurtosis | -0.131 | -0.579 | -0.685 | -0.541 | -0.747 | 1.062 | -0.707 |
| $r_{it}$ | 0.543 | 0.470 | 0.544 | 0.533 | 0.516 | 0.518 | 0.599 |
| $a_{iid}$ | 0.771 | 0.785 | 0.770 | 0.772 | 0.776 | 0.776 | 0.760 |

* $p < .05$.

** $p < .01$.

*** $p < .001$.

Lower triangle for Spearman correlations; upper triangle for Pearson correlations; $r_{it}$ = Corrected item-total correlations; $a_{iid}$ = Cronbach's alpha, if item deleted.

## Factor structure and comparison with other PS constructs

The factor analysis results for sample 2 (n = 1,259) replicate the findings of sample 1. The KMO test and Bartlett's test of sphericity values are 0.827 and chi-square = 2229.075 ($p <$ .001), respectively. The newly proposed PS-7 records 60.716% of the total variance explained by the EFA with oblique rotation. The overreactivity items (6, 10, 14 and 17) have factor loadings ranging from 0.695 to 0.905 and explain 44.614% of the variance. The laxness items (16, 20 and 30) with λ = 0.750 to 0.827 explain 16.102% of the variance. The coefficient alpha of PS-7 (0.790) in sample 2 is also above the acceptable level.

Table 4 shows the CFA results (sample 2; n = 1,259) for the original PS [15] and various shortened versions suggested in the literature [16, 20, 30, 31, 35]. All of the models evaluated in this study are without correlating measurement errors. The CFA results suggest that none of the above scales meet the minimum criteria for adequate or good model fit. The results for the original PS scale are $\chi^2$ (4979.560) / 294 = 16.94, SRMR = 0.086 and RMSEA = 0.113. The other four shortened versions [20, 30, 31, 35] also fail to obtain a satisfactory model fit, with either the $\chi^2$/df or RMSEA values being too low. The CFA results for the latest PS-8 version proposed by Kliem, Lohmann [16] satisfies all of the cut-off values for good fit other than $\chi^2$ / df > 3.

The CFA results indicate that PS-7 has good model fit, with $\chi^2$ (21.809) / 13 = 1.68, $p$ = 0.058, SRMR = 0.020, CFI = 0.999, TLI = 0.998 and RMSEA = 0.023. The standardised factor loadings for the CFA results are high, ranging from 0.64 to 0.77. Overall, the results indicate that PS-7 generally has good fit for a two underlying factor structure without any post hoc modifications.

## Construct validity

This section further evaluates the psychometric properties of PS-7 with reference to the construct validity based on the data from samples 2 (n = 1,259) and 3 (n = 1,259). The CFA results in Table 5 (see Fig 1 for estimated model) suggest that all of the models fulfil the criteria for good model fit. In particular, the results for sample 3 (α = 0.79; ω = 0.84) are $\chi^2$ (16.729) / 13 = 1.29, $p$ = 0.212, SRMR = 0.017, CFI = 0.999, TLI = 0.999 and RMSEA = 0.015. The results support the two-factor structure of PS-7, i.e., laxness (items 16, 20 and 30) and overreactivity (items 6, 10, 14 and 17).

## Criterion validity and discriminant validity

Table 6 shows that PS-7 is strongly correlated with the original PS-26 in terms of the total score and the subscales, namely overreactivity and laxness, for the entire sample (N = 3,777). PS-7 is very significantly positively correlated ($r$ = 0.916, $r_s$ = 0.915, $p < 0.001$) with PS-26 and

**Table 4. Confirmatory factor analysis of the parenting scale.**

| Model [No. of factor/item] | $\chi^2$ | df | $\chi^2$/df | RMSEA [90% CI] | CFI | TLI | SRMR |
|---|---|---|---|---|---|---|---|
| Aronld et al., (1993) [3/26] | 4979.560 | 294 | 16.94 | 0.113 [0.110–0.116] | 0.958 | 0.953 | 0.086 |
| Salari et al., (2012) [2/21] | 2277.121 | 188 | 12.11 | 0.094 [0.091–0.098] | 0.975 | 0.972 | 0.069 |
| Rhoades & O'Leary, (2007) [3/13] | 554.963 | 62 | 8.95 | 0.080 [0.074–0.086] | 0.987 | 0.983 | 0.053 |
| Irvine et al., (1999) [2/12] | 254.697 | 53 | 4.81 | 0.055 [0.048–0.062] | 0.994 | 0.992 | 0.038 |
| Reitman et al., (2001) [2/10] | 265.459 | 34 | 7.81 | 0.074 [0.066–0.082] | 0.988 | 0.984 | 0.047 |
| Kliem et al., (2019) [2/8] | 73.870 | 19 | 3.89 | 0.048 [0.037–0.060] | 0.995 | 0.993 | 0.031 |
| PS-7 [2/7] | 21.809 | 13 | 1.68 | 0.023 [0.000–0.040] | 0.999 | 0.998 | 0.020 |

**Table 5. Factor loadings and fit indices in confirmatory factor analysis for the PS-7, by sample (see Fig 1 for estimated model).**

| Factor/question number | | Sample 2 | | | Sample 3 | | | Combo | | |
|---|---|---|---|---|---|---|---|---|---|---|
| | | Mother | Father | All | Mother | Father | All | Mother | Father | All |
| Laxness (LAX) | | | | | | | | | | |
| 16 | $\lambda_1$ | 0.653 | 0.607 | 0.640 | 0.655 | 0.609 | 0.630 | 0.654 | 0.606 | 0.635 |
| 20 | $\lambda_2$ | 0.726 | 0.727 | 0.737 | 0.774 | 0.832 | 0.726 | 0.750 | 0.780 | 0.732 |
| 30 | $\lambda_3$ | 0.821 | 0.760 | 0.768 | 0.800 | 0.805 | 0.800 | 0.810 | 0.785 | 0.784 |
| Overreactivity (OVE) | | | | | | | | | | |
| 6 | $\lambda_4$ | 0.706 | 0.701 | 0.686 | 0.684 | 0.733 | 0.705 | 0.694 | 0.718 | 0.695 |
| 10 | $\lambda_5$ | 0.679 | 0.664 | 0.695 | 0.662 | 0.681 | 0.672 | 0.671 | 0.674 | 0.683 |
| 14 | $\lambda_6$ | 0.718 | 0.764 | 0.701 | 0.714 | 0.767 | 0.736 | 0.716 | 0.765 | 0.719 |
| 17 | $\lambda_7$ | 0.754 | 0.767 | 0.775 | 0.792 | 0.777 | 0.760 | 0.773 | 0.773 | 0.767 |
| Latent factor covariance | | | | | | | | | | |
| Laxness ~ Overreactivity | $\phi_{l,o}$ | 0.614 | 0.623 | 0.638 | 0.638 | 0.612 | 0.615 | 0.627 | 0.618 | 0.627 |
| Model fit | | | | | | | | | | |
| N | | 735 | 524 | 1,259 | 735 | 524 | 1,259 | 1,470 | 1,048 | 2,518 |
| RMSEA | | 0.000 | 0.025 | 0.023 | 0.002 | 0.039 | 0.015 | 0.020 | 0.041 | 0.024 |
| RMSEA 90% CI | | 0.000–0.030 | 0.000–0.053 | 0.000–0.040 | 0.000–0.037 | 0.008–0.064 | 0.000–0.034 | 0.000–0.036 | 0.025–0.057 | 0.013–0.035 |
| SRMR | | 0.018 | 0.025 | 0.020 | 0.019 | 0.039 | 0.017 | 0.018 | 0.026 | 0.017 |
| $\chi^2$ (df = 13) | | 10.331 | 17.219 | 21.809 | 13.048 | 23.135 | 16.729 | 20.593 | 35.322 | 31.736 |
| $\chi^2$/df | | 0.79 | 1.32 | 1.68 | 1.00 | 1.78 | 1.29 | 1.58 | 2.72 | 2.44 |
| CFI | | 0.999 | 0.999 | 0.999 | 0.999 | 0.998 | 0.999 | 0.999 | 0.997 | 0.999 |
| TLI | | 0.999 | 0.998 | 0.998 | 0.999 | 0.996 | 0.999 | 0.999 | 0.996 | 0.998 |

Combo = sample 2 plus sample 3 (n = 2,518)

its sub-scales, i.e., laxness ($r = 0.830$, $r_s = 0.817$, $p < 0.001$) and overreactivity ($r = 0.850$, $r_s = 0.852$, $p < 0.001$).

The results presented in Table 7 replicate the relationship between PS-7 and the other construct-related scales suggested in the literature [30, 31, 35]. The CBC aggressive and delinquent dimensions are significantly moderately correlated with PS-7 and the laxness and overreactivity subscales. The parents' reports on reactive and proactive aggression are also positively correlated with PS-7, with $r = 0.318$ ($p < 0.001$) and $r = 0.249$ ($p < 0.001$), respectively. PS-7 is

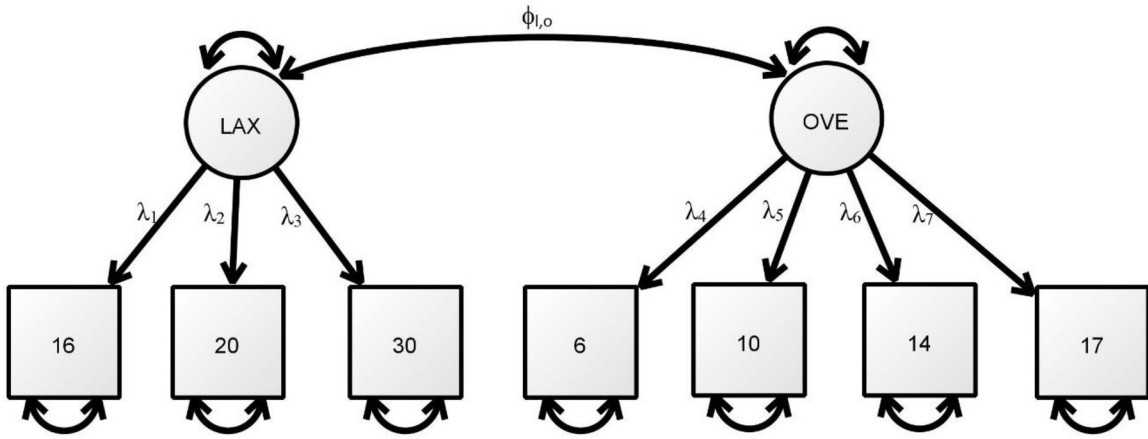

**Fig 1. Estimated model of the 7-item parenting scale.**

**Table 6. Correlations for the PS-7 and PS-26 sub-scales (N = 3,777).**

| Scale | (1) | (2) | (3) | (4) | (5) | (6) |
|---|---|---|---|---|---|---|
| 1. PS-7: total score | 1.000 | 0.814*** | 0.885*** | 0.916*** | 0.830*** | 0.850*** |
| 2. PS-7: laxness | 0.804*** | 1.000 | 0.450*** | 0.714*** | 0.878*** | 0.462*** |
| 3. PS-7: overreactivity | 0.885*** | 0.453*** | 1.000 | 0.836*** | 0.573*** | 0.937*** |
| 4. PS-26: total score | 0.915*** | 0.700*** | 0.844*** | 1.000 | 0.863*** | 0.878*** |
| 5. PS-26: laxness | 0.817*** | 0.877*** | 0.566*** | 0.844*** | 1.000 | 0.589*** |
| 6. PS-26: overreactivity | 0.852*** | 0.467*** | 0.940*** | 0.887*** | 0.584*** | 1.000 |

* $p < .05$.

** $p < .01$.

*** $p < .001$.

Lower triangle for Spearman correlations; upper triangle for Pearson correlations

also significantly correlated with authoritarian parenting styles, such as physical coercion ($r = 0.383$, $p < 0.001$), punitive behaviour ($r = 0.399$, $p < 0.001$) and verbal hostility ($r = 0.495$, $p < 0.001$). The parents reported that their children manifested emotional and behavioural symptoms, including ADHD ($r = 0.354$, $p < 0.001$), oppositional behaviour ($r = 0.388$, $p < 0.001$), cognitive problems ($r = 0.315$, $p < 0.001$) and hyperactivity ($r = 0.355$, $p < 0.001$). This also correlates with the shortened version of the PS. The results also replicate the finding that PS-7 is not significantly related to the educational level of the parent [31], with the results showing that $r = -0.009$ ($p = 0.581$), PS-7: laxness $r = -0.016$ ($p = 0.333$) and PS-7: overreactivity $r = 0.001$ ($p = 0.963$). Thus, PS-7 generally has good criterion and divergent validity.

## Discussion

The main contribution of this study is to introduce PS-7, a shortened version of the original PS. PS-7 and its sub-scales have very strong significantly positive relationships with the

**Table 7. Correlations between the PS-7 in relation to other construct-related scales (N = 3, 777).**

| Scale | PS-7 | PS-7: Laxness | PS-7: Overreactivity |
|---|---|---|---|
| Criterion validity | | | |
| CBC: Aggressive | 0.347*** | 0.220*** | 0.358*** |
| CBC: Delinquent | 0.304*** | 0.183*** | 0.320*** |
| RPQ-parent-report: reactive | 0.318*** | 0.195*** | 0.335*** |
| RPQ-parent-report: proactive | 0.249*** | 0.154*** | 0.259*** |
| PSDQ: physical coercion | 0.383*** | 0.163*** | 0.456*** |
| PSDQ: punitive | 0.399*** | 0.204*** | 0.450*** |
| PSDQ: verbal hostility | 0.495*** | 0.223*** | 0.581*** |
| Parent rating: ADHD | 0.354*** | 0.228*** | 0.360*** |
| Parent rating: Oppositional | 0.388*** | 0.243*** | 0.402*** |
| Parent rating: Cognitive problem | 0.315*** | 0.196*** | 0.327*** |
| Parent rating: Hyperactivity | 0.355*** | 0.219*** | 0.368*** |
| Divergent validity | | | |
| Parent's educational level | -0.009 | -0.016 | -0.001 |

* $p < .05$.

** $p < .01$.

*** $p < .001$.

original scale and its sub-scales, which suggests that PS-7 is comparable to the original PS. The proposed scale retains the original two-factor structure, i.e., laxness and overreactivity, as suggested in the PS literature [16, 20, 31, 35]. The shortened scale also demonstrates good psychometric properties in terms of internal consistency and factorial, criterion and discriminant validity. Thus, the proposed PS-7 provides a handy instrument for researchers and practitioners wishing to evaluate parenting practices for fathers and mothers of young adolescents.

PS-7 is preferable to the existing versions for the following reasons. First, the adapted scale has no complicated items and possesses better factorial validity, with the CFA results suggesting an excellent model fit. Second, in some studies, only EFA and Cronbach's alpha were used to evaluate the metrics of the scales [15, 20]. In this study, the proposed PS-7 was subjected to a series of rigorous tests and comprehensive psychometric tools were used to develop and validate the scale. The results showed that PS-7 has a better factor structure than and is comparable to the original PS. Finally, PS-7 does not rely on correlating the error terms to fulfil the stringent requirements of the goodness-of-fit in CFA. Nonetheless, the proponents of the existing PS versions largely relied on modification indices to improve the model fit [16, 18, 30, 35]. According to Hermida [82], it is inappropriate to allow correlated errors in SEM without strong theoretical justification. Hence, PS-7 is more favourable than the existing PS versions.

Some PS items were not included in PS-7 mainly due to concerns about cultural sensitivity and the contextual rules and regulations. The notion of paternalism is deeply embedded in Asian societies [83, 84]. Therefore, item 12 (*When I want my child to stop doing something, I coax or beg my child to stop*) and item 21 (*If saying "No" doesn't work, I offer my child something nice so he/she will behave*) are less likely to be relevant in an Asian context when parents interact with their children. Similarly, the scenario in item 22 (*When my child misbehaves, I get so frustrated or angry that my child can see I'm upset*) is unlikely to arise in Chinese society because the notion of *face* prevents parents from showing any signs of weakness in front of their children [85]. In many societies, including Hong Kong, laws and regulations forbid parents imposing physical punishment and leaving their children unattended at home [86]. Therefore, item 15 (*When we're not at home, I let my child get away with a lot more*) and item 18 (*When my child misbehaves, I spank, slap, grab, or hit my child*) may not be applicable in those societies. Future studies should consider the significance of such cultural differences.

A potential limitation of this study is that only a limited number of construal-related scales were used to evaluate the criterion validity of PS-7. In the PS literature, the scales are normally cross-checked with measures such as depression, anxiety, self-esteem, confidence, parent-child relationship, impulsivity and social support [16, 30, 31, 35]. Due to the availability of Chinese validated scales and to avoid a lengthy questionnaire, this study used other well-developed scales related to children's aggressive and delinquent behaviour, authoritative parenting, ADHD and oppositional, cognitive and hyperactivity symptoms, which have been extensively discussed and used in the PS literature. The sample used in this study may also limit the generalisability of the findings given that the respondents were recruited from junior secondary schools in Hong Kong and the lack of any evaluation of test-retest reliability. However, these limitations may have been compensated by the large sample size and inclusion of father and mother respondents. Further research is needed to replicate our findings or apply PS-7 in other contexts, preferably with cross-cultural longitudinal research designs in different societies, and ideally involving fathers and mothers of children of different ages.

## Conclusions

To sum up, parenting plays a vital role in child development. There is an urgent need for a shorter and more reliable measure to evaluate different parenting styles and the effectiveness

of parental intervention programmes. The results of this study suggested that the proposed PS-7 had a better factor structure and psychometric properties than the original and other shortened versions of the PS. PS-7 also possesses good internal consistency and criterion validity, with the results being comparable to those for the full version of the PS. The seven-item version of the PS can provide a cost-effective method for assessing parenting practices and conducting epistemological surveys.

## Supporting information

**S1 Dataset.**
(ZIP)

**S1 Appendix. Factor structure of parenting scale and different shortened versions.**
(DOCX)

## Acknowledgments

The authors would like to thank Prof. Lawrence Gerstein for commenting on an early draft of the work. All errors and mistakes are the responsibility of the authors.

## Author Contributions

**Conceptualization:** Sai-fu Fung, Annis Lai Chu Fung.

**Data curation:** Annis Lai Chu Fung.

**Formal analysis:** Sai-fu Fung.

**Funding acquisition:** Annis Lai Chu Fung.

**Investigation:** Sai-fu Fung.

**Methodology:** Sai-fu Fung.

**Project administration:** Annis Lai Chu Fung.

**Resources:** Annis Lai Chu Fung.

**Supervision:** Annis Lai Chu Fung.

**Validation:** Sai-fu Fung.

**Writing – original draft:** Sai-fu Fung.

**Writing – review & editing:** Annis Lai Chu Fung.

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
