## [Decision Letter · Decision Letter 0]

30 Aug 2019

PONE-D-19-17632

Development and evaluation of the psychometric properties of a brief parenting scale (PS-7) for the parents of adolescents

PLOS ONE

Dear Dr. Fung,

Thank you for submitting your manuscript to PLOS ONE. After careful consideration, we feel that it has merit but does not fully meet PLOS ONE’s publication criteria as it currently stands. Therefore, we invite you to submit a revised version of the manuscript that addresses the points raised during the review process.

This manuscript seeks to refine the Parenting Scale into a brief form and examine associations with an additional measure of parenting, parental education, and youth symptoms. I was able to have the input of two experts in the area and I thank them for their constructive, but critical input on the work. I highlight a number of concerns that I shared with the reviewers and note several additional elements of the work that would need to be addressed in a revision.

Both reviewers and I note that the conceptual foundation for the work was fairly modest. As the PS was developed with young children, more needs to be stated about how the measure translates to adolescence.

Most critically, Reviewer 1 and I share the concern that no details are provided about how the short measure was developed. What criteria were employed to select the items? How many iterations of items were tested before arriving at the particular item set? The scientific processes that lead to the items and, ultimately, the measure is paramount. Without pre-registration, it becomes very challenging to identify serendipity in selecting the items, or a clear rational process.

Reviewer 1 also notes that additional statistical information is needed to evaluate the bifactor model.

Reviewer 2 noted that there was additional attention needed to fathers. I concur with the statement and press the issue further. In order to make stronger claims about fathers, additional analyses may be needed to clarify whether mothers and fathers provide similar or dissimilar responses. Moreover, it was not clear whether only one parent per household was included, or if there were some parent dyads who both completed the measures. If the latter, more attention in the analyses may be needed to accommodate non-independence of observations.

This is a fairly extensive set of comments to address. I cannot be sure that your work will ultimately be accepted for publication. However, there is interest in the work and the data come from an impressive sample. 

We would appreciate receiving your revised manuscript by Oct 14 2019 11:59PM. To enhance the reproducibility of your results, we recommend that if applicable you deposit your laboratory protocols in protocols.io, where a protocol can be assigned its own identifier (DOI) such that it can be cited independently in the future. For instructions see: http://journals.plos.org/plosone/s/submission-guidelines#loc-laboratory-protocols

We look forward to receiving your revised manuscript.

Kind regards,

Thomas M. Olino

Academic Editor

PLOS ONE

Journal Requirements:

Reviewers' comments:

Reviewer's Responses to Questions

**Comments to the Author**

1. Is the manuscript technically sound, and do the data support the conclusions?

Reviewer #1: Yes

Reviewer #2: Yes

2. Has the statistical analysis been performed appropriately and rigorously? 

Reviewer #1: No

Reviewer #2: Yes

3. Have the authors made all data underlying the findings in their manuscript fully available?

Reviewer #1: No

Reviewer #2: Yes

4. Is the manuscript presented in an intelligible fashion and written in standard English?

Reviewer #1: No

Reviewer #2: Yes

5. Review Comments to the Author

Reviewer #1: This manuscript focuses on assessing the psychometric properties of a brief, 7-item version of the parenting scale for use with parents of adolescents. Strengths of this paper include the assessment of psychometric properties of a parenting practices measure to be used with adolescents, as well as the utilization of a diverse population. Nevertheless, I do have several concerns.

1.    There are several typos and misspellings. Please proofread carefully, as these errors make reading the manuscript difficult. Further, the manuscript appears to jump from topic to topic without transition, which makes reading the manuscript difficult. Please review the manuscript for overall flow.

2.    The explaining for why this scale should be used with parents of adolescents is not clearly outlined. Please provide a more thorough explanation, as the original measure was designed for use with parents of young children.  The manuscript was presented in a way that suggested the emphasis was on use with parents of adolescents; however, this is not discussed later within the manuscript. Please clarify what the primary goal of the manuscript is and why it is a significant contribution.

3.    Additionally, there should be a more substantial focus in the Introduction on the use of the PS with Chinese families, and some of the details provided in the Discussion on cultural differences in parenting should be a focal point of the study rationale.

4.    Please include the average age of the children in the study. Additionally, please provide more demographic information.

5.    Please explain what a “complex item” is in more detail. As it currently stands, it is unclear precisely what this means or why it is referred to as “complex.”

6.    Please provide more detail regarding the procedure for creating the brief scale. How did the authors choose these seven items? They differ from other versions, so it is unclear how these items were selected.

7.    In the fourth paragraph of the procedures section, the Discussion regarding the PS and associations with parenting behaviors and child factors appears abruptly and should not be presented in this section. Please move to the rationale to the Introduction and detailed description of each measure in the measures section.

8.    At times throughout the manuscript, it is unclear if the authors are referring to the broader parenting scale literature, or the specific parenting scale they are validating. Please clarify.

9.    It is unclear why it would be essential to reduce the scale from 21 items to 7. Additionally, it was stated that there is already an 8-item scale; therefore, is it necessary to reduce the scale by one item? Please explain. Further, research using IRT has shown that the 21-item version is superior to briefer versions (see Lorber et al., 2014).

10. Given the sample size, the authors should test for measurement invariance across both parent and child gender.

11. In the EFA there appeared to be high cross-loadings. With the 21-item version, could it be the case that a bifactor structure is more appropriate (e.g., lax, overactivity, and overall behavioral control)?

12. Please report both alpha and omega for reliability as well as 95% CIs for reliability (available in several R packages).

13. Lastly, a significant limitation of the current study is relying on only a single informant and not including test-retest reliably.

Reviewer #2: The submitted manuscript proposed an abbreviated version of the Parenting Scale (PS) for parents of adolescents based on a sample recruited from Hong Kong. This manuscript contributes to the parenting literature and measurement of parenting by indicating a shorter and more accessible form of an already well validated assessment of parenting. The authors used rigorous statistical and research methods and reported their findings clearly and concisely. This manuscript is a welcomed addition to the literature as it is an in-depth analysis of a parenting measure using an Asian sample, which is significant given the noted cultural differences in parenting practices with regard to paternalism as discussed by the authors. Further, I commend the author’s analysis of an existent parenting measure created narrowly for mothers of younger children and expanding these factors with a sample consisting of parents of adolescents including both mothers and fathers.

Along with the strengths of the manuscript discussed above, there were some areas of weakness that might strengthen its contribution to the literature. First, the authors discussed issues of validity as well as the problems with the statistical rigor of previous iterations of the PS in great detail but do not sufficiently state their theoretical basis for their aims and specific hypotheses based on said theory. The authors bring up many important variables that require further investigation in the measurement of parenting, such as lacking attention given to possible differences in parenting styles and behaviors with regard to child age (e.g. adolescents), historical failure to include diverse samples with regard to parent gender, as well as important cultural considerations when applying these measures globally. However, the manuscript would be improved by further discussion of hypotheses of how these items/factors may vary when studying only parents of adolescents. Similarly, consideration of how the inclusion of fathers may influence or improve these analyses and resultant proposed measure (e.g. do fathers parents differently, interaction of parent gender with the examination of adolescence and/or Asian culture). Discussion of differences in parenting in Asian cultures was addressed in the discussion section, but only to substantiate why items were removed from the outset. More thought and analysis of how these results fit into these multicultural concepts is warranted and an important addition to the literature.

6. PLOS authors have the option to publish the peer review history of their article (what does this mean?). If published, this will include your full peer review and any attached files.

Reviewer #1: No

Reviewer #2: Yes: Raelyn Loiselle

---

## [Author Response · Author response to Decision Letter 0]

16 Oct 2019

Comments from Editor:

1. Both reviewers and I note that the conceptual foundation for the work was fairly modest. As the PS was developed with young children, more needs to be stated about how the measure translates to adolescence.

Response: We have addressed this concern in the revised manuscript with additional discussion in the background section on the Parenting Scale to support the scale is applicable to both father and mother of early adolescent (p. 2). 

2. Most critically, Reviewer 1 and I share the concern that no details are provided about how the short measure was developed. What criteria were employed to select the items? How many iterations of items were tested before arriving at the particular item set? The scientific processes that lead to the items and, ultimately, the measure is paramount. Without pre-registration, it becomes very challenging to identify serendipity in selecting the items, or a clear rational process. 

Response: We have responded it in the Reviewer 1 comment (point number 6). In addition, we also added the limitations and future research direction on p. 14.

3. Reviewer 1 also notes that additional statistical information is needed to evaluate the bifactor model 

Response: Additional EFA (with different estimators and rotation methods) was conducted with the 21-item version. The results did not support the bifactor structure.

4. Reviewer 2 noted that there was additional attention needed to fathers. I concur with the statement and press the issue further. In order to make stronger claims about fathers, additional analyses may be needed to clarify whether mothers and fathers provide similar or dissimilar responses. Moreover, it was not clear whether only one parent per household was included, or if there were some parent dyads who both completed the measures. If the latter, more attention in the analyses may be needed to accommodate non-independence of observations.

Response: We also addressed this concern in Reviewer 2 (point number 2). We computed additional CFA on both mother and father samples. The results are identical to the samples consist of both parents (p. 11, Table 5).

There was only one parent per household was included in this study.

5. This is a fairly extensive set of comments to address. I cannot be sure that your work will ultimately be accepted for publication. However, there is interest in the work and the data come from an impressive sample.

Response: Thank you for inviting us to resubmit the manuscript. We have carefully considered all issues raised by the Editor, Reviewer 1 and Reviewer 2 and have made all edits that have been suggested.

Comments from Reviewer 1:

Reviewer #1: This manuscript focuses on assessing the psychometric properties of a brief, 7-item version of the parenting scale for use with parents of adolescents. Strengths of this paper include the assessment of psychometric properties of a parenting practices measure to be used with adolescents, as well as the utilization of a diverse population. Nevertheless, I do have several concerns. 

Response: Many thanks for your comments and feedbacks. We have made substantial edits to the manuscript to address your concerns.

1. There are several typos and misspellings. Please proofread carefully, as these errors make reading the manuscript difficult. Further, the manuscript appears to jump from topic to topic without transition, which makes reading the manuscript difficult. Please review the manuscript for overall flow. 

Response: The revised manuscript has been re-organised the structure and proofread carefully.

2. The explaining for why this scale should be used with parents of adolescents is not clearly outlined. Please provide a more thorough explanation, as the original measure was designed for use with parents of young children. The manuscript was presented in a way that suggested the emphasis was on use with parents of adolescents; however, this is not discussed later within the manuscript. Please clarify what the primary goal of the manuscript is and why it is a significant contribution. 

Response: The Parenting Scale has been validated and used on parents of early adolescent. We have added the relevant literature in the introduction section (p. 2).

3. Additionally, there should be a more substantial focus in the Introduction on the use of the PS with Chinese families, and some of the details provided in the Discussion on cultural differences in parenting should be a focal point of the study rationale. 

Response: Agreed. In the revised manuscript, we added some relevant literature related to the Chinese parenting style (p. 2).

4. Please include the average age of the children in the study. Additionally, please provide more demographic information. 

Response: Our study only asked the parent’s age and other demographic information is presented in Table 1 (p. 4-5).

5. Please explain what a “complex item” is in more detail. As it currently stands, it is unclear precisely what this means or why it is referred to as “complex.” 

Response: Change to complicated, i.e. the item is related to more than one dimension/factor.

6. Please provide more detail regarding the procedure for creating the brief scale. How did the authors choose these seven items? They differ from other versions, so it is unclear how these items were selected. 

Response: In the revised manuscript, we have clarified the item selection procedure in p. 7.

In the discussion section (p. 13-14), we further illustrated that the brief scale with better psychometric properties and culturally universal, especially suitable for the Chinese context. 

7. In the fourth paragraph of the procedures section, the Discussion regarding the PS and associations with parenting behaviors and child factors appears abruptly and should not be presented in this section. Please move to the rationale to the Introduction and detailed description of each measure in the measures section. 

Response: The discussion regarding the PS and associations with parenting behaviours and child factors in the methods section is to explain and justify the scales/variables for evaluating the criterion validity of the Parenting Scale. If move it to the introduction, it may affect the flow of the manuscript. But we do agree to provide detailed description of each measure in this section (p. 6).

8. At times throughout the manuscript, it is unclear if the authors are referring to the broader parenting scale literature, or the specific parenting scale they are validating. Please clarify. 

Response: The PS literature reviewed/quoted are all related to the specific parenting scale derived from Arnold et al. (1993).

9. It is unclear why it would be essential to reduce the scale from 21 items to 7. Additionally, it was stated that there is already an 8-item scale; therefore, is it necessary to reduce the scale by one item? Please explain. Further, research using IRT has shown that the 21-item version is superior to briefer versions (see Lorber et al., 2014). 

Response: The original Parenting Scale consist of 26 items with three factor structure. The 21-item version is a shortened version proposed by Salari et al. (2012) and evaluated by Lorber et al. in 2014. However, the existing Parenting Scale literature in recent years suggested that a briefer version is possessing better psychometric properties, such as the latest 8-item version.

This study also evaluated different versions with empirical data. Table 4 (p. 9) summarised the results of original scale (26 items with three factor structure), as well as the 21-item and other shortened versions. The CFA results show that none of them fulfilled the criteria for model fit. Hence, this study attempts to develop and validate a parenting style that can suitable the cultural context of the Chinese society.

Also, in the discussion section, some items are removed from the original scale mainly due to the concerns of the cultural/legal context in the Chinese society (p. 14).

10. Given the sample size, the authors should test for measurement invariance across both parent and child gender. 

Response: We computed additional CFA analysis of the PS-7 on both father and mother respondents. The results also suggested that the proposed scale with good model fit (p. 11, Table 5). However, we did not collect the child gender, as this variable is usually not reported in other parenting scale studies.

11. In the EFA there appeared to be high cross-loadings. With the 21-item version, could it be the case that a bifactor structure is more appropriate (e.g., lax, overactivity, and overall behavioral control)?

Response: Additional EFA (with different estimators and rotation methods) was conducted with the 21-item version. The results did not support the bifactor structure.

12. Please report both alpha and omega for reliability as well as 95% CIs for reliability (available in several R packages). 

Response: We have computed the alpha and omega for reliability with 95% CIs. Relevant texts are added in the methods and results sections (p. 6, 7 and 8).

13. Lastly, a significant limitation of the current study is relying on only a single informant and not including test-retest reliably. 

Response: The sample was randomly stratified into three datasets based on the sex ratio of the respondents to avoid any potential biased of the analysis. We do agree that without test-retest reliability is a limitation of this study, we have added this in the discussion section (p. 14).

Comments from Reviewer 2:

Reviewer #2: The submitted manuscript proposed an abbreviated version of the Parenting Scale (PS) for parents of adolescents based on a sample recruited from Hong Kong. This manuscript contributes to the parenting literature and measurement of parenting by indicating a shorter and more accessible form of an already well validated assessment of parenting. The authors used rigorous statistical and research methods and reported their findings clearly and concisely. This manuscript is a welcomed addition to the literature as it is an in-depth analysis of a parenting measure using an Asian sample, which is significant given the noted cultural differences in parenting practices with regard to paternalism as discussed by the authors. Further, I commend the author’s analysis of an existent parenting measure created narrowly for mothers of younger children and expanding these factors with a sample consisting of parents of adolescents including both mothers and fathers. 

Response: Thanks for the comments and feedbacks. 

In the revised manuscript, we have provided additional discussion in the background section on the Parenting Scale to support the scale is applicable to both father and mother of early adolescent. We also took your advice to discuss the Asian parenting style (p. 2).

Along with the strengths of the manuscript discussed above, there were some areas of weakness that might strengthen its contribution to the literature. First, the authors discussed issues of validity as well as the problems with the statistical rigor of previous iterations of the PS in great detail but do not sufficiently state their theoretical basis for their aims and specific hypotheses based on said theory. The authors bring up many important variables that require further investigation in the measurement of parenting, such as lacking attention given to possible differences in parenting styles and behaviors with regard to child age (e.g. adolescents), historical failure to include diverse samples with regard to parent gender, as well as important cultural considerations when applying these measures globally. However, the manuscript would be improved by further discussion of hypotheses of how these items/factors may vary when studying only parents of adolescents. Similarly, consideration of how the inclusion of fathers may influence or improve these analyses and resultant proposed measure (e.g. do fathers parents differently, interaction of parent gender with the examination of adolescence and/or Asian culture). Discussion of differences in parenting in Asian cultures was addressed in the discussion section, but only to substantiate why items were removed from the outset. More thought and analysis of how these results fit into these multicultural concepts is warranted and an important addition to the literature. 

Response: We shared your concerns about these issues, the following are the changes we made it in the revised manuscript:

In the background section, we highlighted the importance of cultural considerations for the parenting scale (p. 2).

In the results section, we provided additional clarification of the development of the 7-item PS (p. 7).

We computed additional CFA on both mother and father samples. The results are identical to the samples consist of both parents (p. 11, Table 5).

In the discussion section, we proposed future study may consider incorporating diverse samples in different societies ideally with different parent gender and parent with different child age (p. 14).

Comments from Editorial Office:

1. We note that you have indicated that data from this study are available upon request. PLOS only allows data to be available upon request if there are legal or ethical restrictions on sharing data publicly. For more information on unacceptable data access restrictions, please see http://journals.plos.org/plosone/s/data-availability#loc-unacceptable-data-access-restrictions. 

Response: We have uploaded the data set as Supporting Information file through the online submission system. It can enables the future readers to replicate our findings.

2. Please ensure that you refer to Figure 1 in your text as, if accepted, production will need this reference to link the reader to the figure.

Response: Thank you for the reminder. We have added Figure 1 in the text (p. 9).

3. Thank you for including your ethics statement on the online submission form: 

"This study was approved by the ethical committee of the City University of Hong Kong. Its procedure was in compliance with the Declaration of Helsinki guidelines. All of the participants gave informed consent prior to the study.". 

To help ensure that the wording of your manuscript is suitable for publication, would you please also add this statement at the beginning of the Methods section of your manuscript file.

Response: We have added the ethics statement at the beginning of the Method section (p. 4).

---

## [Editor Report · Decision Letter 1]

26 Nov 2019

PONE-D-19-17632R1

Development and evaluation of the psychometric properties of a brief parenting scale (PS-7) for the parents of adolescents

PLOS ONE

Dear Dr. Fung,

Thank you for submitting your manuscript to PLOS ONE. After careful consideration, we feel that it has merit but does not fully meet PLOS ONE’s publication criteria as it currently stands. Therefore, we invite you to submit a revised version of the manuscript that addresses the points raised during the review process.

Unfortunately, I was unable to obtain comments from the initial reviewers of the manuscript. However, I read the revision and responses to previous comments carefully. Overall, this manuscript is improved. Though, there continues to be an important limitation in the description of how the short form items were selected. The paper describes administering the full 26 item measure. What analyses were conducted on these data to reduce the items from 26 to 7? Without this information there is no scientific evidence supporting the decision. There are additional description in the text, but there are no details provided about how those recommendations were used in this study.

We would appreciate receiving your revised manuscript by Jan 10 2020 11:59PM. To enhance the reproducibility of your results, we recommend that if applicable you deposit your laboratory protocols in protocols.io, where a protocol can be assigned its own identifier (DOI) such that it can be cited independently in the future. For instructions see: http://journals.plos.org/plosone/s/submission-guidelines#loc-laboratory-protocols

We look forward to receiving your revised manuscript.

Kind regards,

Thomas M. Olino

Academic Editor

PLOS ONE

---

## [Author Response · Author response to Decision Letter 1]

6 Jan 2020

Comments from Editor:

Unfortunately, I was unable to obtain comments from the initial reviewers of the manuscript. However, I read the revision and responses to previous comments carefully. Overall, this manuscript is improved. Though, there continues to be an important limitation in the description of how the short form items were selected. The paper describes administering the full 26 item measure. What analyses were conducted on these data to reduce the items from 26 to 7? Without this information there is no scientific evidence supporting the decision. There are additional description in the text, but there are no details provided about how those recommendations were used in this study.

Responses from Authors:

Thank you for your comments. In the revised manuscript, we have added a section to clarify the item selection process (p. 5 and 6):

The process is based on the criteria, latest practice and recommendations used in the existing PS studies (16, 18) and other scale development and validation literature (42-49). The selected items have gone through the following two-step procedure. Step one, selecting the items: i) using inductive approach to analyze the correlation matrix of all the items and keeping the items with 0.250 or above. We also cross-checking the Cronbach’s alpha, if deleted and McDonald’s omega values to ensure that the shortened version is above the acceptable range > 0.70; ii) using scree test in factor analysis to identify the factor structure with eigenvalues higher than 1.0 (50). We also select the items with highest factor loadings, i.e. > .50 and avoid items involve correlating the error terms based on the modification indices. When selecting the items, we try to retain the sufficient items (at least three) in each factor to ensure that the validity standard of the shortened version is equivalent to the full version; iii) to verify the abbreviated version with the confirmatory factor analysis to ensure that the scale with good construct validity, i.e. fulfil all the stringent requirements for good model fit. Step two, ensuring that the compatibility between the full scale and shortened version: iv) we adopted the following practice of Kliem, Lohmann (16), ‘short form should also correlate strongly with the original PS on the total score level as well as on the subscale (overreactivity and laxness) level’ (p. 34). As such, there should be significant strong positive correlation (> 0.80) between the full and short scales, including their sub-scales; and v) lastly, the abbreviated version should possessing good criterion validity as reported in the existing PS literature.

We also made additional changes in the results (p. 7 and 8) and references sections (p. 19):

42. Loewenthal KM. An introduction to psychological tests and scales. 2 ed: Philadelphia, Pa. : Psychology Press; 2001.

43. Schel SHH, Bouman YHA, Vorstenbosch ECW, Bulten BH. Development of the forensic inpatient quality of life questionnaire: short version (FQL-SV). Quality of Life Research. 2017;26(5):1153-61.

44. MacKenzie MB, Kocovski NL, Blackie RA, Carrique LC, Fleming JE, Antony MM. Development of a Brief Version of the Social Anxiety - Acceptance and Action Questionnaire. Journal of Psychopathology and Behavioral Assessment. 2017;39(2):342-54.

45. Markos A, Kokkinos CM. Development of a short form of the Greek Big Five Questionnaire for Children (GBFQ-C-SF): Validation among preadolescents. Personality and Individual Differences. 2017;112:12-7.

46. Smith GT, McCarthy DM, Anderson KG. On the sins of short-form development. Psychological Assessment. 2000;12(1):102-11.

47. Svedholm-Hakkinen AM, Lindeman M. Actively open-minded thinking: development of a shortened scale and disentangling attitudes towards knowledge and people. Think Reasoning. 2018;24(1):21-40.

48. Chae D, Park Y. Development and Cross-Validation of the Short Form of the Cultural Competence Scale for Nurses. Asian Nurs Res. 2018;12(1):69-76.

49. Zhang XT, Wang MC, He LN, Jie L, Deng JX. The development and psychometric evaluation of the Chinese Big Five Personality Inventory-15. Plos One. 2019;14(8):21.

50. Cattell RB. The Scree Test For The Number Of Factors. Multivariate Behavioral Research. 1966;1(2):245-76.

---

## [Editor Report · Decision Letter 2]

8 Jan 2020

PONE-D-19-17632R2

Development and evaluation of the psychometric properties of a brief parenting scale (PS-7) for the parents of adolescents

PLOS ONE

Dear Dr. Fung,

Thank you for submitting your manuscript to PLOS ONE. After careful consideration, we feel that it has merit but does not fully meet PLOS ONE’s publication criteria as it currently stands. Therefore, we invite you to submit a revised version of the manuscript that addresses the points raised during the review process.

Thank you for the additions to the manuscript. In my previous comments, I requested information about the decision making process to lead to the selected 7 items. The steps are described in the revision. However, the specific application of the steps is not detailed. There are two potential remedies. One would be to add the empirical information that led to the elimination of the items into the manuscript. An alternative would be to provide the complete data and step-by-step analytic syntax as supplementary material that would permit a reader to follow the work. I leave the decision to you to take one of the approaches.

We would appreciate receiving your revised manuscript by Feb 22 2020 11:59PM. To enhance the reproducibility of your results, we recommend that if applicable you deposit your laboratory protocols in protocols.io, where a protocol can be assigned its own identifier (DOI) such that it can be cited independently in the future. For instructions see: http://journals.plos.org/plosone/s/submission-guidelines#loc-laboratory-protocols

We look forward to receiving your revised manuscript.

Kind regards,

Thomas M. Olino

Academic Editor

PLOS ONE

---

## [Author Response · Author response to Decision Letter 2]

9 Jan 2020

Comments from Editor:

Thank you for the additions to the manuscript. In my previous comments, I requested information about the decision making process to lead to the selected 7 items. The steps are described in the revision. However, the specific application of the steps is not detailed. There are two potential remedies. One would be to add the empirical information that led to the elimination of the items into the manuscript. An alternative would be to provide the complete data and step-by-step analytic syntax as supplementary material that would permit a reader to follow the work. I leave the decision to you to take one of the approaches.

Responses from Authors:

Many thanks for your comments and recommendations for the two potential remedies. We would like to use the second way, i.e. to provide the complete data and all the analytic syntax (in both SPSS and R) that we used as supplementary material. We have uploaded those files on the Editorial Manager.

---

## [Editor Report · Decision Letter 3]

13 Jan 2020

Development and evaluation of the psychometric properties of a brief parenting scale (PS-7) for the parents of adolescents

PONE-D-19-17632R3

Dear Dr. Fung,

We are pleased to inform you that your manuscript has been judged scientifically suitable for publication and will be formally accepted for publication once it complies with all outstanding technical requirements.

With kind regards,

Thomas M. Olino

Academic Editor

PLOS ONE
---

## [Editor Report · Acceptance letter]

16 Jan 2020

PONE-D-19-17632R3 

Development and evaluation of the psychometric properties of a brief parenting scale (PS-7) for the parents of adolescents 

Dear Dr. Fung:

I am pleased to inform you that your manuscript has been deemed suitable for publication in PLOS ONE. Congratulations! Your manuscript is now with our production department. 

With kind regards,

on behalf of

Dr. Thomas M. Olino 

Academic Editor

PLOS ONE